# Use of Nutraceutical Ingredient Combinations in the Management of Tension-Type Headaches with or without Sleep Disorders

**DOI:** 10.3390/nu13051631

**Published:** 2021-05-13

**Authors:** Paolo Curatolo, Romina Moavero

**Affiliations:** 1Child Neurology and Psychiatry Medicine, Systems Medicine Department, Tor Vergata University of Rome, Via Montpellier 1, 00133 Rome, Italy; romina.moavero@opbg.net; 2Child Neurology Unit, Neuroscience Department, Bambino Gesù Children’s Hospital, IRCCS, Piazza S. Onofrio 4, 00165 Rome, Italy

**Keywords:** tension type headache, sleep, insomnia, melatonin, tryptophan, vitamins, nutraceuticals, children

## Abstract

Headache is the most common pain complaint in the pediatric population, with tension type headache (TTH) having a prevalence of 10–15% in children. Up to 70% of pediatric patients with chronic headache also experience sleep disruption, with a likely bidirectional relationship between headache and poor sleep. Treatment options include specific pharmacological approaches as well as non-pharmacological alternatives; nutraceuticals have the advantage of a relative lack of side effects. Exogenous melatonin has been shown to be useful and safe in improving sleep-wake cycles and quality of sleep in children, helping to regulate the circadian rhythm, with a secondary positive impact on headache. Supplementation with other nutraceutical ingredients, such as tryptophan, magnesium, and B vitamins, can have significant additional effects in children with primary headache, with or without sleep disorders. Tryptophan may reduce night awakenings and improve the efficiency of sleep. Primary headache has been related to low amounts of magnesium in serum, and integration with magnesium appears to be effective in reducing headache attacks without adverse effects. There are different observational reports and uncontrolled studies suggesting a possible synergistic effect for these nutraceuticals, but there is now a need for high-quality randomized controlled trials in order to confirm these positive preliminary findings.

## 1. Introduction

Primary headache disorders are prevalent and disabling conditions that can affect people of all ages, including children and adolescents, with a significant impact on school-related and social activities as well as quality of life [1,2]. Headache is the most common pain complaint in the pediatric population, with more than 50% of children and adolescents suffering from headache at some point in their lifetime [3]. Tension type headache (TTH) presents with bilateral mild-to-moderate pain, without nausea and phono- or photophobia [4]. It is usually an episodic condition with a prevalence ranging from 10 to 15% in children. Recurrent TTH may cause school and behavioral problems. A recent study on a wide sample of pediatric patients applying the International Headache Society criteria confirmed that patients with TTH presented significant emotional difficulties [5]. Another recent population-based study showed that TTH was associated with decreased health-related quality of life [6].

Different studies on the pathophysiology of TTH suggest an altered production of nociceptive substances playing a role in the genesis of pain and pericranial muscle tenderness, but genetic predisposition also seems to play a crucial role [7,8]. Some reports suggest a possible role for serotonin (5-hydroxytryptamine, 5-HT)—which is a neutrotransmitter of the endogenous analgesic system and limbic system—in exercising some serotoninergic control of the pain threshold [9,10,11]. Moreover, plasmatic tryptamine levels appear to be very low in subjects with chronic TTH, thus reinforcing a possible role of this system in the pathogenesis of this disorder [12].

Many children and adolescents with headache also experience sleep disruption, and there is a likely bidirectional relationship between headache and poor sleep, with headache negatively impacting sleep, and vice versa [13], with about 70% of pediatric patients with chronic headache also experiencing sleep difficulties [14]. The combination of headache and poor sleep may be particularly disruptive to the functioning of children and adolescents. Sleep disorders have been observed to be more prevalent in patients with migraine compared to healthy controls [15]. Children and adolescents with chronic headache usually identify poor sleep as a primary trigger for their headache [16]. Existing conceptual models on the relationship between sleep and headache in children and adolescents propose different possible connections, but the mechanism of action remains largely unknown. A meta-analysis revealed that the presence of insomnia increased the risk of headache, and also of chronic pain worsening the long-term prognosis [17].

Treatment options include specific pharmacological approaches as well as non-pharmacological alternatives [18]. Nutraceuticals have the advantage of a relative lack of side effects.

Up to now the number of clinical studies on TTH in children and adolescents is very limited.

In this study we review the current role of non-pharmacological treatments, including melatonin, tryptophan, and magnesium, which have the potential to help reduce the burden associated with TTH in children and adolescents.

## 2. Melatonin and TTH

Melatonin is crucially involved in the regulation of our circadian rhythm, being secreted by the pineal gland during darkness and suppressed by light [19]. It interacts with MT1 and MT2 receptors (melatonin receptors 1 and 2), involved in REM (Rapid Eye Movements) and NREM (Non-REM) sleep, respectively, and through their activation it also regulates the vigilance states. Melatonin is characterized by strong chronobiotic as well as hypnotic properties, and when administered exogenously at the right time and the right dosage, it can modulate the sleep-wake rhythm [20,21,22]. In particular, melatonin presents the best efficacy for sleep onset insomnia, being able to significantly reduce sleep onset latency. Some evidence also suggests that it may increase total sleep time [23]. Besides this very important function, melatonin also plays a role in many other biological processes, showing antioxidant and anti-inflammatory effects, and participating in free radical scavenging [22]. Furthermore, melatonin is also able to ameliorate weight gain, reducing body fat accumulation and, thus, diminishing the deleterious effects of obesity and its inflammatory profile [24]. Melatonin exerts its multiple actions by interacting with several receptors, including the opioid, benzodiazepine, muscarinic, nicotinic, serotonergic, α1-adrenergic, and α2-adrenergic receptors at multiple central nervous system levels [22].

Therefore, melatonin not only has a role in sleep disorders, but can also play a role in headache pathophysiology through several mechanisms [23]. Indeed, it has antinociceptive effects on both inflammatory and neuropathic pain [25] (Figure 1). Melatonin also modulates cytokines, interleukins, and TNF-alpha (tumor necrosis factor alpha) [22]. It is likely that its action of restoring the circadian rhythms also enhances adaptive capabilities, thus playing a role in chronic pain management [17,26]. Finally, melatonin also shows an anxiolytic effect that may be important in treating chronic pain conditions [27,28].

Melatonin’s chronobiological effect may be able to reduce headache in patients with circadian rhythm sleep disorders and headache. In particular, adequately timed and dosed melatonin treatment ameliorated headache in 78.6% of 328 patients, while it was able to induce a slight headache in 13.8% of 676 patients with circadian rhythm sleep disorders without headache [29].

Circadian sleep-wake dysregulation and sleep disorders are strongly associated with primary headache disorders, and sleep dysregulation has been reported in TTH [30].

In a systematic review and meta-analysis study, Liampas et al. showed that melatonin may be of potential benefit in the treatment and prevention of migraine attacks in adults, but evidence from a high-quality randomized controlled trial is still required [31]. Still less data are available for TTH, and in particular for pediatric TTH. In an uncontrolled study, 12 adult patients with TTH were enrolled and administered 4 mg melatonin, reporting a statistically significant decrease in both headache frequency and Headache Impact Test score [32].

A study on 61 adult patients with TTH showed that a treatment with 3 mg melatonin resulted in a decrease in the number of headaches per month as well as in pain intensity; furthermore, there was also an improvement in quality of life, with a reduction of the scores in the questionnaires for depression and anxiety [26].

A 3-month open label trial enrolled a total of 22 children with primary headache. Among them, 8 presented with TTH, and were treated 3 mg melatonin [33]. Half of them displayed a reduction of at least 50% in the frequency of headache attacks, but none of them presented a complete remission of symptoms [33].

Melatonin is usually safe—the most common adverse events include daytime sleepiness, headache, dizziness, and hypothermia. Very rare adverse events are considered to be serious, and no life-threatening reactions have been reported. Most side effects are self-limited, or immediately resolve after withdrawing medication [34].

Studies on melatonin use in primary headache have had some limitations, such as lack of adequate control and placebo, and overall they have tended to have a small sample size.

## 3. Tryptophan/5-Hydroxytryptophan and TTH

Tryptophan, or L-tryptophan (Trp/5-HTP), is an essential amino acid crucial for the synthesis of several molecules, including serotonin, melatonin, tryptamine, and kynuramines [35]. Tryptophan is metabolized via two basic pathways: the indole–kynuramine–niacin pathway, and the serotonin–melatonin pathway [36,37]. The vast majority of tryptophan is metabolized and degraded through the kynurenine pathway generating neuroactive molecules, which act as antagonists of NMDA (N-methyl-D-aspartate) receptors [38]. 5-Hydroxytryptophan (5-HTP) is the intermediate metabolite between Trp and serotonin, and is able to cross the blood–brain barrier; when administered exogenously it increases the biosynthesis of serotonin in the central nervous system (CNS) [39,40]. In the CNS, serotonin levels have been implicated in the regulation of sleep, depression, anxiety, aggression, appetite, temperature, sexual behavior, and pain sensation [41]. The exact mechanisms through which Trp/5-HTP act on sleep are multiple and not completely clarified. However, it is clear that they increase the CNS concentration of serotonin, provide substrates for melatonin production, and enhance serotonin-mediated regulation of sleep [42].

Therapeutic administration of 5-HTP has been shown to be effective in treating a wide variety of conditions. A randomized double-blind study enrolled 78 patients with TTH treated with 5-HTP 3 times a day for 8 weeks; the follow-up period was prolonged for 2 further weeks after withdrawing treatment. All patients kept a diary reporting headaches, severity of pain, and use of analgesics [43]. Treatment with 300 mg per day 5-HTP showed a significant decrease in the consumption of analgesics in comparison with the placebo group, as well as a significant decrease in the number of days with headache, without any significant adverse effects [43].

5-HTP might be associated with mild nausea, especially in the first phase of treatment, but this effect is usually transient [41,44]. When prescribing 5-HTP it is important to keep in mind that if taken with a selective serotonin reuptake inhibitor it might cause a serotonin syndrome—characterized by agitation, confusion, delirium, tachycardia, diaphoresis, and blood pressure alterations [41].

## 4. Magnesium and TTH

Magnesium is crucial for numerous enzyme reactions, and is involved in energy metabolism. A large body of evidence suggests that low levels of magnesium may lead to neuronal dysfunction, and are often observed in individuals with headache [45]. Magnesium also acts as an antagonist of calcium channels, preventing the excessive activation of excitatory synapses such as NMDA (N-methyl-d-aspartate) receptors, and plays a role in downregulating inflammation [46]. Furthermore, it may block serotonin-induced vasoconstriction in blood vessels [46].

Recent evidence suggests a relationship between magnesium deficiency and mild-to-moderate TTH [46,47,48] (Figure 2). Due to its high potential bioavailability, magnesium may have special relevance for the treatment of some neurological conditions, including headache, and could be a well-tolerated option for reducing the frequency of attacks, up to dosages of 1500 mg [46].

In a matched case–control study patients with headache showed lower serum levels of magnesium during and between migraine attacks compared to healthy individuals [49]. A recent study suggests inadequate magnesium consumption intake is associated with headache in the United States [50].

From a tolerability point of view, magnesium appears to be absolutely safe, with most of the studies reporting either no adverse events, or only soft stools or diarrhea [46].

## 5. B vitamin Supplementation and TTH

B vitamins act as coenzymes in a great proportion of the enzymatic processes that underpin every aspect of cellular physiological findings, including generation of energy and anabolic metabolism. Vitamin B6 acts as a cofactor in the synthesis of several neurotransmitters, including melatonin and serotonin [51].

A systematic literature search evaluating studies assessing the supplementation of B6 and B12 to migraine patients showed a potential beneficial effect with an attractive safety profile [52]. Dosages used in the different studies analyzed by Liampas et al. have been widely variable: up to 300 mg for vitamin B6, and up to 400 mcg for vitamin B12. Recent evidence in children suggested that those with TTH presented serum levels of vitamin B12 significantly lower than age-matched healthy controls [53].

A 40-week randomized double-blind trial evaluated children aged 6–13 years with migraine and TTH treated with riboflavin compared with a placebo group [54]. Although no significant benefits were observed on migraine, riboflavin significantly reduced the frequency of attacks in children with TTH [54].

### 5.1. The Effect of Melatonin, Magnesium, and Vitamin B Complex Supplementation in the Treatment of Insomnia

A randomized study enrolled 60 patients suffering with insomnia randomly assigned either to a control group or to treatment with a magnesium–melatonin–vitamin B complex (175 mg liposomal magnesium oxide, 10 mg vitamin B6, 16 mcg vitamin B12, 1 mg melatonin). This supplementation for 3 months appeared to be highly effective for insomnia, especially in reducing the number of night awakenings, regardless of its aetiology; it also ameliorated the daytime consequences of sleep disturbance, thus improving quality of life [55].

### 5.2. The Effect of Melatonin, Tryptophan, and Vitamin B6 Supplementation on Chronic Headache

A recent study administering 3 mg melatonin, 60 mg tryptophan, and 4.2 mg vitamin B6 for 2 months to children with chronic headache, compared with children receiving 3 mg melatonin alone, enrolled a total of 34 children with headache [56]. Overall, 90% of the enrolled children presented an improvement of their headache in the 2 months of nutraceutical supplementation, with benefits already observed in the first month, and similar results for children taking melatonin alone and those taking melatonin plus tryptophan and vitamin B6. In terms of night awakenings, an overall improvement was reported in 78.8% of children, with a statistically significant benefit in the group additionally supplemented with tryptophan and vitamin B6 [56].

In this small observational trial of children with primary headache treated with melatonin, tryptophan, and vitamin B6 supplementation, an improvement of headache attacks was observed already after 4 weeks of treatment.

These findings should be considered preliminary until these treatment options are assessed in larger, multicenter studies with longer durations. This study had some limitations, and further trials will be needed to define factors that predict responses to treatment and to determine the optimal dosage of nutraceuticals for TTH.

## 6. Conclusions

Primary headache disorders can be severe and disabling, with a negative impact on functioning and quality of life of children. Many children and adolescents with chronic headaches also experience sleep difficulties, and there is a likely bidirectional relationship between chronic headache and sleep problems.

Exogenous melatonin has been shown to be useful and safe in improving sleep–wake cycles and quality of sleep in children. Tryptophan may reduce night awakenings and improve the efficiency of sleep. Primary headache has been related to low levels of magnesium in serum, and integration with magnesium appears to be effective in reducing headache attacks without adverse effects. Therefore, melatonin may help to regulate the circadian rhythm, with a secondary positive impact on headache, and supplementation with other nutraceutical ingredients—such as tryptophan, magnesium and B vitamins—may have significant additional effects in children with primary headache, with or without sleep disorders. There are different observational reports and uncontrolled studies suggesting a possible synergistic effect for these nutraceuticals, but there is now a need for high-quality randomized controlled trials in order to confirm these positive preliminary findings.

## Figures and Tables

**Figure 1 nutrients-13-01631-f001:**
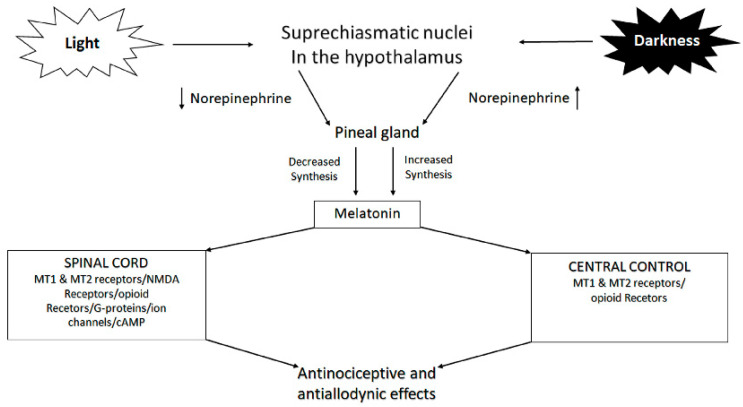
Control of melatonin synthesis in the pineal gland by light-dark cycles; melatonin’s antinociceptive and antiallodynic effects. Abbreviations: NMDA: N-methyl-D-aspartate, cAMP: Cyclic adenosine monophosphate; MT1: melatonin receptor 1; MT2: melatonin receptor 2.

**Figure 2 nutrients-13-01631-f002:**
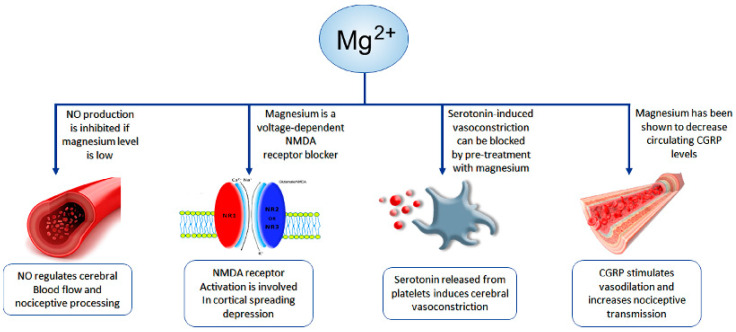
Mechanisms involved in migraine, and the possible role of magnesium. CGRP, circulating calcitonin gene-related peptide; NMDA, N-methyl-D-aspartate; NO, nitric oxide.

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
