# Peer review of "Use of Nutraceutical Ingredient Combinations in the Management of Tension-Type Headaches with or without Sleep Disorders"

_nutrients, 2021, doi:10.3390/nu13051631_

Round 1

Reviewer 1 Report

This is a great review. I would've liked to see more data about the sleep onset insomnia or total sleep time in subjects with treatments and control. Did you have any actigraphy data on these patients? 

If there is availability to make a table comparing the different studies with different treatment options and if available levels of nutrients ( Vit B6, B12, tryptophan, and mg). Otherwise, if not accessible, more controlled studies with specific nutrients levels pre, and post-treatment and length of treatment are required.

I agree with you more studies are needed esp multicenter controlled studies.

Thank you!

Author Response

Thank you for this review. We fully agree with Reviewer that much more data would be useful to have a more complete picture of the role of nutraceuticals in this significant comorbidity between headache and sleep disorders. However, the literature lacks of rigorous papers focused both on headache and sleep disorders, so that when headache is the main focus usually data on sleep are scarce. Actigraphy data were not available for these studies, nor nutrients levels. So, a table was not possible due to paucity of data, but we added some detail of a study focusing also on sleep at page 5, lines 195-6.

Reviewer 2 Report

This review focuses or summarizes and quantifies evidence about the prevention effects of melatonin, tryptophan (& analogs), and magnesium and vitamins B on children with primary headache with and without sleep disorders (with a detrimental impact on children's performance and overall quality of life). Where fifteen review - study were included in the “analysis”. The authors provide or summarize the main findings that the beneficial effects of mentioned supplements on patients with primary headache with and without sleep disorders.

Mostly seem to be 2 observational trials of children, one with insomnia randomly assigned and another a small observational trial of children with primary headache. Both treated with melatonin and supplemented with tryptophan and vitamin B6. In addition, give excellent implications for future research “Longer follow-up of larger randomized trials in larger multicenter studies is necessary to clarify the benefits of melatonin treatment and tryptophan and vitamin B6 supplementation and/or Mg nutrition supplements in children with primary headache with and without sleep disorders. These observations together with its high safety profile of both, melatonin drug and tryptophan, magnesium, and vitamins B supplementation nutrition make at least the above-mentioned treatment and supplements a potentially useful tool for a stand-alone or adjunct therapy for children with primary headache with and without sleep disorders. Which I find all interesting to the readers.

The very short review has smooth writing, as well as accurate and innovative Figures. I recommend acceptance

A small comment

On pages 2 and lines 78-79, the authors should add the role of melatonin administration in metabolism, especially the positive anti-obesogenic effect, since obesity in children is more and more frequent and can be related to sleep disturbance, headache, and migraine. 

More reference:

Barrea L, et all. The clock diet: a practical nutritional guide to manage obesity through chrononutrition. Minerva Med. 2021 Apr 29. doi: 10.23736/S0026-4806.21.07207-4. Epub ahead of print. PMID: 33913659.

Tarantino S, et all. Anxiety, Depression, and Body Weight in Children and Adolescents With Migraine. Front Psychol. 2020 Oct 28;11:530911. doi: 10.3389/fpsyg.2020.530911. PMID: 33192771; PMCID: PMC7655930.

Fernández Vázquez G, et all. Melatonin increases brown adipose tissue mass and function in Zücker diabetic fatty rats: implications for obesity control. J Pineal Res. 2018 May;64(4):e12472. doi: 10.1111/jpi.12472. Epub 2018 Mar 25. PMID: 29405372.

Author Response

Thank you very much, a sentence has been added (lines 80-82)